# Impact of Plant-Based School Meals on Gut *Bifidobacterium* spp. Abundance and Health Outcomes in Schoolchildren from Bahia, Brazil

**DOI:** 10.3390/ijms252313073

**Published:** 2024-12-05

**Authors:** Camilla Almeida Menezes, Dalila Lucíola Zanette, Letícia Bastos Magalhães, Renata Maria Rabello da Silva Lago, Alexvon Nunes Gomes, Ronald Alves dos Santos, Ana Marice Teixeira Ledeia, Nelzair Araújo Vianna, Ricardo Riccio Oliveira

**Affiliations:** 1Instituto Gonçalo Moniz, Fundação Oswaldo Cruz, Fiocruz, Salvador 40296-710, Bahia, Brazil; camilla.almeida@fiocruz.br (C.A.M.); letybastosm@gmail.com (L.B.M.); ronald.santos@fiocruz.br (R.A.d.S.); nelzair.vianna@fiocruz.br (N.A.V.); 2Instituto Carlos Chagas, Fundação Oswaldo Cruz, Fiocruz, Curitiba 81350-010, Paraná, Brazil; dalila.zanette@fiocruz.br; 3Escola Bahiana de Medicina e Saúde Pública, Salvador 40290-000, Bahia, Brazil; lagorenata1973@gmail.com (R.M.R.d.S.L.); angomes3@bahiana.edu.br (A.N.G.); anamarice@bahiana.edu.br (A.M.T.L.)

**Keywords:** *Bifidobacterium* spp., plant-based diet, school-based intervention, gut microbiota, public health

## Abstract

Plant-based diets have been linked to various health benefits, including an improved gut microbiota composition, potentially influencing non-communicable diseases. This study investigates the impact of a school meal intervention on the gut microbiota, specifically the abundance of *Bifidobacterium* spp. (BIF), in Brazilian schoolchildren. A quasi-experimental intervention was conducted in 2019 across four municipalities in the semi-arid region of Bahia, Brazil. The Sustainable School Program aimed to replace animal-based and ultra-processed foods with plant-based options. Clinical, dietary, anthropometric, and laboratory data were collected at the beginning and end of the school year. Fecal samples were analyzed for BIF abundance using RT-PCR. The intervention improved anthropometric and laboratory outcomes, including increased serum hemoglobin levels and reduced LDL-cholesterol. Despite these benefits, no significant change in BIF abundance was observed. However, a negative correlation between BIF abundance and waist-to-height ratio was found. While the intervention positively affected several health parameters, it did not significantly alter BIF abundance. Nevertheless, the abundance of BIF may explain some of these positive outcomes. The findings highlight the potential of plant-based diets to improve overall health, but suggest that further research is needed to understand the role of the gut microbiota in these outcomes. Future studies should explore the influence of factors such as physical activity on the gut microbiota and health.

## 1. Introduction

The relationship between plant-based diets and health outcomes has been investigated. Meta-analysis results suggest protective effects against type 2 diabetes [1], dyslipidemia [2], cardiovascular diseases [3], various types of digestive cancers [4], premature death [5], and all-cause mortality [6]. These effects seem to be associated with the ability to beneficially modulate the gut microbiome [7].

Several factors can modify the composition and function of the gut microbiota, with dietary patterns being one of the most influential [8]. The consumption of plant-based foods, with their higher fiber and phytochemical content, appears to be more conducive to healthy gut microbiota, with a more significant presence of *Bifidobacterium* and *Prevotella* genera [9]. This effect is attributed to the specific metabolites they produce, such as short-chain fatty acids, membrane proteins, and 5′-methylthioadenosine [10]. Conversely, the consumption of animal-based foods has been associated with unfavorable outcomes, with a higher presence of bile-resistant bacteria such as *Bilophila* and *Bacteroides* [11,12]. Additionally, the consumption of ultra-processed foods can negatively alter the gut microbiota composition due to changes in micronutrient density and energy content, as well as the presence of food additives and advanced glycation end-products resulting from the thermal processing of these foods [13].

An imbalance in the microbial population, with a predominance of pathogenic bacteria over probiotic ones, is characterized as intestinal dysbiosis [14]. The primary consequence of dysbiosis is an increased intestinal mucosal permeability, which can lead to endotoxemia caused by lipopolysaccharides, an increase in pro-inflammatory cytokine synthesis, macrophage infiltration into adipose tissue, and subsequent local and systemic inflammation [15]. This is key in triggering peripheral insulin resistance, obesity, metabolic syndrome, and cardiovascular diseases [16,17]. The gut microbiota profile associated with obesity shows a higher prevalence of Firmicutes, fewer Bacteroidetes, and a lower relative proportion of *Bifidobacterium* spp. [18].

Obesity has significantly increased in the Brazilian population, including among schoolchildren [19,20,21,22]. The health conditions of school-aged children can be significantly influenced by their school, which is a potential environment for health promotion, as well as nutritional, environmental, and humanitarian education [23]. School-based interventions tend to positively influence students’ eating behaviors, particularly nutrition-related ones, with a greater availability of fruits, vegetables, and legumes [24,25]. Brazil’s National School Feeding Program, the country’s oldest and largest public policy on food and nutrition security, represents a valuable tool for promoting health and environmental, economic, and social sustainability [26].

The Public Prosecutor’s Office of Bahia, Brazil, proposed an initiative to change the school menus in four state municipalities [27]. The Sustainable School Program (SSP) aimed to reduce the provision of animal-based and ultra-processed foods in school meals while increasing the availability of locally produced vegetables. The project involved approximately 150 schools and over 32,000 students.

Studies assessing sustainability promotion initiatives through sustainable menus typically focus on the nutritional and environmental aspects of menus and their acceptability among students [28,29,30,31]. Data on the health effects of these initiatives on this specific population are limited. Similarly, studies that examine the composition of the gut microbiota as a result of interventions are typically carried out in controlled environments during clinical trials. This study aims to investigate this topic within the context of public policy. Its objectives are to evaluate the effects of the SSP intervention on schoolchildren’s health and to determine the role of *Bifidobacterium* spp. in these results.

## 2. Results

Among 124 participants, the average age was 9.6 years (±2.9), with 58.1% attending schools in urban areas. Of the students, 53.2% were male, 64.5% were delivered vaginally, and 91.9% were breastfed. However, only 52.4% received exclusive breastfeeding until the sixth month of life, and 70.2% had used antibiotics before the age of five. Food consumption data, anthropometry, and laboratory findings were assessed at the beginning (D0) and the end (D240) of the 2019 school year. The results of paired analyses are presented below.

Regarding the participants’ food consumption (Figure 1), significant increases were observed in total calorie intake (1746 Kcal vs. 1854 Kcal, *p* = 0.038), protein (228 Kcal vs. 271 Kcal, *p* = 0.003), total fat (577 Kcal vs. 689 Kcal, *p* = 0.006), saturated fat (12.5 Kcal vs. 13.6 Kcal, *p* = 0.024), cholesterol (113 g/1000 Kcal vs. 140 g/1000 Kcal, *p* = 0.004), and unprocessed foods (643 Kcal vs. 722 Kcal, *p* = 0.007). Fiber intake was also investigated, but there was no statistically significant difference in consumption after the intervention (7.4 g/1000 Kcal vs. 7.6 g/1000 Kcal, *p* = 0.645).

The paired analysis of anthropometric parameters revealed increases in height (*p* < 0.0001), weight (*p* < 0.0001), and waist circumference (*p* = 0.005). However, a reduction in WHtR was observed. These data are available in Figure 2.

Among the laboratory findings presented in Figure 3, the increase in serum hemoglobin levels (*p* < 0.0001) and the reduction in LDL-cholesterol (*p* < 0.0001) stand out. Similarly, there was an increase in triglyceride levels (*p* < 0.0001) and a decrease in vitamin D (*p* < 0.0001). There was no change in ferritin and vitamin B12 levels nor the intestinal abundance of *Bifidobacterium* spp. between the two time points investigated. Participants with a vitamin B12 deficiency at D0 were referred for treatment and excluded from this analysis. According to Table 1, this scenario decreased the frequency of low hemoglobin from 11% to 2% (*p* = 0.007) and hypercholesterolemia from 14% to 6% (*p* = 0.049).

Although no change in the intestinal abundance of bifidobacteria was observed, since there were changes in the anthropometric and laboratory parameters, the relationship between the abundance of these bacteria and these findings was investigated. The results of a Spearman correlation analysis are shown in Figure 4. A weak but significant negative correlation (r = −0.2464, *p* = 0.036) was observed between the gut abundance of *Bifidobacterium* spp. and WHtR.

Table 2 presents the results from a Generalized Estimating Equations (GEEs) analysis, with a Gaussian family and an independent correlation structure, to evaluate the impact of the school meal intervention and *Bifidobacterium* spp. intestinal abundance on the clinical outcomes. The model was adjusted for potential confounders, including sex, age, and school location. Estimated coefficients (ECs) are reported with standard errors (SEs) and *p*-values. The findings suggest that the school meal intervention was negatively associated with WHtR (EC = −0.010, *p* = 0.023) and serum LDL-cholesterol levels (EC = −7.488, *p* = 0.008) and positively associated with serum hemoglobin levels (EC = 0.752, *p* < 0.0001). However, the intervention appeared to increase serum triglyceride levels (EC = 17.041, *p* < 0.0001). Regarding the impact of the gut microbiota on these findings, it was observed that bifidobacteria was negatively associated with blood glucose (EC = −0.00068, *p* = 0.043) and serum LDL-cholesterol levels (EC = −0.00203, *p* = 0.012).

## 3. Discussion

This study evaluated the impact of a school meal program on the physical measurements and laboratory results of 124 students from four municipalities in the semi-arid region of Bahia, Brazil. The students were assessed at the following two time points: the beginning (D0) and the end (D240) of the 2019 school year. The primary focus of the study was to investigate the effect of *Bifidobacterium* spp. intestinal abundance on these outcomes.

A previous study conducted by this research group described that this population had a high intake of sugar, saturated fat, and ultra-processed foods before the school meal intervention. Additionally, there was a high frequency of overweight, cardiovascular risk, vitamin B12 deficiency, hypercholesterolemia, and hypertriglyceridemia [32]. These data suggest that urgent interventions for better access to healthy food were necessary.

The characteristics of the school feeding intervention were assessed and published in another study by the same research group [33]. In summary, it was found that, aside from a decrease in greenhouse gas emissions, there was no significant nutritional difference between plant-based menus and the menus commonly provided by schools, except for some micronutrients. However, the “sustainable” menu showed a more promising nutritional profile in terms of meeting fiber requirements while adhering to the maximum cholesterol and sodium intake limits, which aligns with the intervention’s objective of preventing non-communicable diseases (NCDs) in the school population.

A comparative analysis of the participants before and after the intervention revealed significant findings. Regarding food consumption, increases in total calories, protein, total fat, saturated fat, cholesterol, and unprocessed food intake were observed. It is important to note that the food consumption evaluation presented here was conducted through the 24-h recall method. However, this instrument should be applied at three different non-consecutive time points, representing two typical days and one atypical food consumption [34]. Due to logistical issues, the method could not be applied three times in this study; thus, the presented data represent food consumption on the day before the interview rather than habitual consumption patterns. Although accepted by the scientific literature, interpretations based on these findings should consider this limitation.

The longitudinal assessment of anthropometric indicators showed an increased height, weight, and waist circumference, typical findings in a population experiencing growth and structural development. However, a reduction in WHtR was observed, indicating growth progression with a reduced cardiovascular risk for this population. Similar findings were reported in a meta-analysis that assessed children and adolescents aged from 3 to 18 years exposed to primarily plant-based diets, particularly among those already obese [35].

Regarding laboratory indicators, serum hemoglobin levels increased. Some plant-based menus contain more iron than conventional ones [33]. However, since no significant increase in ferritin status was observed in this population, the rise in hemoglobin levels may also be attributed to factors beyond diet, including the gut microbiota [36].

Another relevant finding was the reduction in serum vitamin D levels, although this was insufficient to represent an increase in the prevalence of a deficiency in this nutrient, which remained low in this population. It is possible that the serum vitamin D levels were higher at D0 due to increased sun exposure from outdoor activities, which are more frequent during the summer vacation. Conversely, after 200 days of the school year, with consequent reductions in the frequency of such activities and sun exposure, serum vitamin D levels are expected to decrease, as they are directly dependent on endogenous synthesis resulting from ultraviolet exposure [37].

A limiting nutrient in plant-based diets is the vitamin B12, as it is only naturally found in animal-based foods, and in children and adolescents, this dietary pattern is associated with nutrient deficiency [38]. Despite excluding animal-based foods from school meals throughout the 2019 school year, there was no change in the serum vitamin B12 levels among participants, likely due to the presence of this vitamin in fortified plant-based foods. It is also worth noting the presence of animal-based foods in meals that the students consumed outside of school, which may have contributed to maintaining their serum vitamin B12 levels. Another relevant finding in this population is the high prevalence of vitamin B12 deficiency, despite the omnivorous dietary pattern.

A significant reduction in serum LDL-cholesterol levels was also observed. Since the intervention aimed to reduce the availability of cholesterol-rich foods, but the dietary intake assessment at D240 revealed an increase in cholesterol intake, it is possible that this increase was due to food consumed outside of school. It is important to remember that the intervention investigated in this study was limited to the meals offered by schools, without interference in other foods consumed inside and outside the school environment. Similarly, as described earlier, the limitations related to the dietary intake assessment conducted in this study must be considered. A meta-analysis of studies investigating children and adolescents showed that exposure to a primarily plant-based diet could also reduce their total cholesterol and LDL-cholesterol levels [39].

Another significant finding refers to the increase observed in triglyceride levels. Although not enough to raise the frequency of hypertriglyceridemia at D240, the presence of this finding was already relevant at D0. Food consumption cannot explain this increase, as the paired analysis did not reveal an increase in carbohydrate intake. Considering the limitations related to the dietary assessment conducted in this study, another explanation for this finding could be the greater availability of carbohydrates and sugar in school meals. However, the assessment of school menus does not support this hypothesis [33]. A relevant aspect not evaluated in this study was physical activity. Sedentary behavior, along with dietary habits, is associated with outcomes such as dyslipidemia, excess weight, and cardiovascular diseases, even in school-age populations [40]. Investigating the practice of physical exercise in this population could help us to understand the outcome related to hypertriglyceridemia. Since 2016, the ERICA study has shown a higher prevalence of hypertriglyceridemia in the northeastern population compared to the rest of the country [41]. Furthermore, the ERICA study demonstrated that the national prevalence of metabolic syndrome was higher among public school students [42]. These data suggest a negative progression in the nutritional and metabolic profile of the Brazilian school-aged population in recent years, highlighting the need for public policies on physical activities and food security to address this situation.

Diets characterized by unprocessed, predominantly plant-based foods and a higher fiber intake, such as the Mediterranean diet, have been associated with a healthier gut microbiome profile. This profile has a greater abundance of *Bifidobacterium* spp. and a reduction in the growth of Firmicutes [43], which has been linked to an improved cardiometabolic health and insulin sensitivity [44]. A study conducted on 294 obese adults aimed to evaluate the effect of dietary patterns on the gut microbiome and the subsequent impact on participants’ health [45]. It was observed that the Mediterranean diet (MED) induced significant changes in the structure of the gut microbiome. However, the plant-based version of the diet (Green-MED) led to more pronounced changes, with a greater diversity of bacterial genera and species. Both diets were associated with gradual improvements in body weight and cardiometabolic biomarkers. However, in this case, the Green-MED diet was associated with a reduction in bifidobacteria, suggesting that other microbiota-related aspects may have accounted for the clinical outcomes, such as the greater abundance of the *Prevotella* genus.

Another previous study conducted with this population at D0 revealed that participants who consumed more meat had a lower gut *Bifidobacterium* spp. abundance [46]. This microbiota profile was also associated with a higher prevalence of hyperglycemia, while a greater bifidobacteria abundance was associated with cardiovascular protection in this population. In this longitudinal study, there was no difference in intestinal *Bifidobacterium* spp. abundance before and after the school meal intervention. It is worth noting that the intervention, initially planned to last for two years, with the frequency of exposure to plant-based menus being four times a week, was interrupted by the COVID-19 pandemic at the beginning of the 2020 school year. Thus, suspending in-person classes due to social distancing requirements reduced the duration and frequency of individuals’ exposure to the proposed intervention, potentially influencing the results. Another factor that may have influenced the lack of significant differences in *Bifidobacterium* spp. abundance after the intervention is that, although the primers used in this study target the 16S ribosomal gene region common to many species within the genus, there is no confirmation that they can detect all currently known species. Nevertheless, a negative correlation was observed between bifidobacteria gut abundance and WHtR, a parameter that was reduced after the intervention.

A GEE analysis was proposed to investigate whether the school meal intervention impacted the observed outcomes and the role of bifidobacteria in these effects. The results, adjusted for confounding variables such as sex, age, and school location (rural or urban), showed that the intervention reduced WHtR and LDL-cholesterol levels and increased hemoglobin. However, it also contributed to a rise in hypertriglyceridemia. An important consideration is that the model interpreted the intervention as the time between D0 and D240. Other dietary factors, in addition to carbohydrate intake, and lifestyle factors, such as physical activity, may have contributed to this outcome. *Bifidobacterium* spp. intestinal abundance, in turn, seemed to contribute less significantly, yet still notably, to reducing blood glucose and LDL-cholesterol levels.

This study demonstrated significant anthropometric and laboratory benefits from the school meal intervention, particularly in reducing cardiovascular risk. These positive outcomes highlight the program’s effectiveness in improving children’s health. While the dietary assessment method and absence of physical activity data may have influenced specific results, such as the increased triglycerides, the overall impact remains clear. Future studies should expand these assessments of dietary patterns and exercise habits and explore interventions with better control over food intake outside the school environment. Additionally, investigating other probiotic components of the gut microbiome could provide further insights. These findings underscore the vital role of public policies in promoting healthy eating habits from an early age.

## 4. Materials and Methods

### 4.1. Study Design

This was a quasi-experimental community intervention study conducted at the public health level. It evaluated implementing a school-based dietary intervention to improve students’ health conditions.

### 4.2. Study Population

Schoolchildren from the cities of Barrocas, Biritinga, Serrinha, and Teofilândia, located in the semi-arid region of the state of Bahia, Brazil, who were enrolled in the municipal public school system in 2019, were invited to participate in the study. One hundred and twenty-four individuals were randomly selected and allocated across 15 schools in both rural and urban areas. Based on the expected effect size of 0.5 [47] and an alpha error probability of 0.05, this sample gave the study a power of 99%. The study included individuals aged between 5 and 19 years with no prior diagnosis of intestinal diseases, food allergies, or intolerances. Students who had used antibiotics in the 30 days preceding stool sample collection were excluded.

### 4.3. Intervention

The SSP, an initiative by the Public Prosecutor’s Office of Bahia, Brazil, aimed to promote public policies focused on improving the quality and quantity of school meals, preventing diseases, optimizing financial and environmental public resources, and promoting nutritional, environmental, and humanitarian education. The program was launched in 2018 and reached 137 schools, 30 daycare centers, and approximately 32,000 students [27].

In 2018, several initiatives were carried out to raise awareness among local leaders and civil society. Technical training was provided for school kitchen staff. At the same time, efforts were made to enhance local family farming and plan improvements to the physical infrastructure of schools, including renovations of kitchen facilities. Educational initiatives focused on nutrition and the environment were also developed, involving students and the broader community. Furthermore, students conducted acceptability tests for meal options to be introduced.

In 2019, the municipalities’ nutritionists responsible for school meals initiated a change in school menus. The plan involved gradually replacing animal-based items with plant-based options, aiming for full implementation by the end of the 2020 school year. The target was to introduce plant-based menus twice a week in 2019, increasing to three times a week in the first half of 2020 and reaching four times a week by the end of 2020. However, the COVID-19 pandemic and the resulting suspension of in-person classes in early 2020 due to social distancing requirements interrupted these planned activities. So, the intervention was carried out in the 2019 school year, as described in Figure 5. The nutritional and environmental characteristics of the menus are detailed in a previous study by this research group [33].

### 4.4. Data Collection

The 124 individuals exposed to the intervention were evaluated at the following two time points: at the beginning (D0) and the end (D240) of the 2019 school year. The evaluation included clinical, dietary, anthropometric, and laboratory data. A trained team collected data in the morning at the school facilities where the participants were enrolled.

#### 4.4.1. Clinical Evaluation

Clinical assessment was conducted through an in-person interview guided by a pre-structured questionnaire. This questionnaire included questions about the individual’s medical history, to be answered by their legal guardian.

#### 4.4.2. Dietary Analysis

Dietary intake was retrospectively assessed using the validated 24-h recall method, which involves identifying and quantifying all foods consumed on the day prior to the interview. A photographic food portion album was used to assist with more accurate portion estimates, allowing the participant or their legal guardian to indicate the portion size of the foods consumed [48]. A quantitative food intake analysis was conducted using the validated Brazilian food composition tables [49,50]. For qualitative analysis, the NOVA food classification was used [51].

#### 4.4.3. Anthropometric Measurements

The anthropometric assessment was based on the Body Mass Index for Age (BMI/A). The participants were dressed in their school uniforms, and their weight was measured using a digital electronic scale (Seca^®^, Hamburg, Germany). Height was measured with the participant barefoot using a portable vertical stadiometer (AVA-312^®^, Três Rios, RJ, Brazil). BMI was calculated using Quetelet’s formula, and BMI/A was classified according to a z-score analysis based on the WHO child growth charts to determine nutritional status [52]. For cardiovascular risk assessment, WC was measured using a non-elastic tape (Balmak^®^, São Paulo, SP, Brazil). WC measurements were compared to the reference curves proposed by Fernández et al. [53]. WHtR was then calculated according to McCarthy and Ashwell [54] and validated for children and adolescents by Nambiar et al. [55]. A WHtR above 0.5 was considered to be indicative of cardiovascular risk.

#### 4.4.4. Laboratory Examination

Blood samples were collected from individuals in a fasting state, and the following indicators were analyzed: complete blood count, blood glucose, total cholesterol and its fractions, triglycerides, ferritin, vitamin B12, and 25-OH vitamin D3. Fecal samples were collected using sterile containers previously provided by the research team. Bacterial DNA extraction was performed using the QIAamp PowerFecal DNA Kit^®^ (QIAGEN, Toronto, ON, Canada). Subsequently, the DNA was quantified, and its purity was verified using a NanoDrop^®^ spectrophotometer (Thermo Fisher Scientific, Waltham, MA, USA). The target microbiota were quantified using the Real-Time Polymerase Chain Reaction (RT-PCR) technique with the Real-time PCR 7500^®^ system (Thermo Fisher Scientific, Waltham, MA, USA) from Fiocruz Technological Platforms Network. The forward ACTCCTACGGGAGGCAGCAG and reverse ATTACCGCGGCTGCTGG primers were used to quantify the total bacteria (TB). The analysis of BIF was performed with the forward GCGTGCTTAACACATGCAAGTC and reverse CACCCTTTCCAGGAGCTATT primers, both provided by Ludwig Biotecnologia^®^. RT-PCR reactions were performed with 5 µL of Sybr Green Master Mix^®^ (Applied Biosystems, Foster City, CA, USA), 0.1 µL of each of the above-mentioned primers, 3.8 µL of ultrapure water, and 1 µL of DNA as template. The amplification protocol followed the following cycle: 50 °C for 2 min, 95 °C for 10 min, followed by 40 cycles of 95 °C for 1 s and 60 °C for 1 min. A Melt Curve step was then performed with the following parameters: 95 °C for 15 s, 60 °C for 1 min, 95 °C for 30 s, and 60 °C for 15 s. The results generated by the equipment’s software are expressed in terms of cycle threshold (Ct) values. From the Ct, the delta Ct (ΔCt) was calculated by subtracting the Ct of the target bacterium (BIF) from the Ct of the reference bacterium (TB). Quantification was expressed in Relative Expression Units (REUs), according to the model by Albesiano et al. [56], by calculating 10,000 divided by 2 raised to the ΔCt. This metric provides a relative estimate of the abundance of BIF compared to the total bacteria present in the sample.

### 4.5. Statistical Analyses

Data tabulation was conducted using the RedCap^®^ platform, with double-entry data input under the supervision of the research coordinators. Statistical analyses were performed using the R^®^ software version 4.1.0 and GraphPad Prism^®^ version 8.2.1. All variables analyzed exhibited am asymmetrical distribution, and their central tendency and dispersion measures are described as the median and interquartile range (IQR). Longitudinal analyses for paired data were conducted using the Wilcoxon test, while comparisons of the proportions in unpaired data were made using Fisher’s exact test. Spearman’s correlation was applied to assess the relationship between intestinal BIF abundance and anthropometric and laboratory outcomes. Generalized Estimating Equations (GEEs) models were used to investigate the effect of the intervention and intestinal BIF abundance on the outcome variables. A *p*-value of less than 0.05 was considered to be statistically significant.

## 5. Conclusions

The school meal intervention significantly improved the laboratory and anthropometric parameters of the studied population, with *Bifidobacterium* spp. gut abundance helping to explain some of these positive outcomes. These results suggest that promoting vegetable consumption and reducing animal-based and ultra-processed foods in school meals through public policies play a crucial role in combating NCDs among children and adolescents in Brazil’s municipal public education system. Future studies should also explore the impact of physical activity and other aspects of the gut microbiome on the health outcomes of this population.

## Figures and Tables

**Figure 1 ijms-25-13073-f001:**
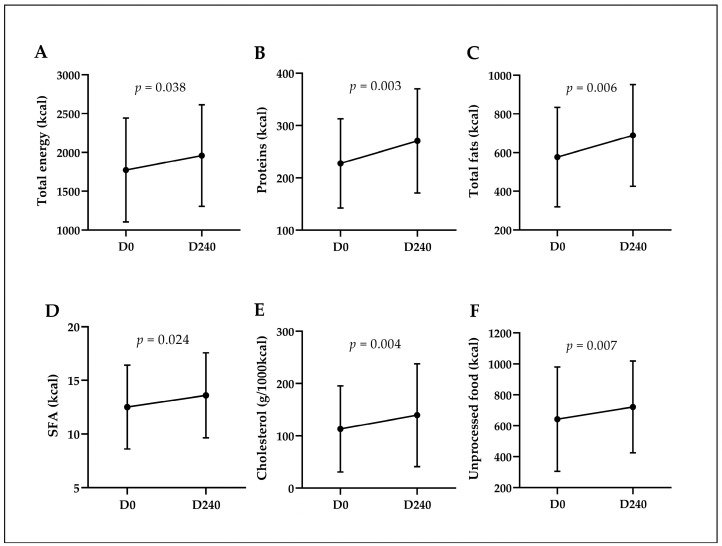
Paired analysis of dietary consumption before and after intervention: statistically significant changes based on the Wilcoxon Test. D0: start of the 2019 academic year; D240: end of the 2019 academic year. SFA: saturated fatty acids. Paired analysis of the consumption of: (**A**) Total calories (Kcal); (**B**) Proteins (Kcal); (**C**) Total fats (Kcal); (**D**) Saturated fat (Kcal); (**E**) Cholesterol (g/1000 Kcal); (**F**) Unprocessed foods (Kcal).

**Figure 2 ijms-25-13073-f002:**
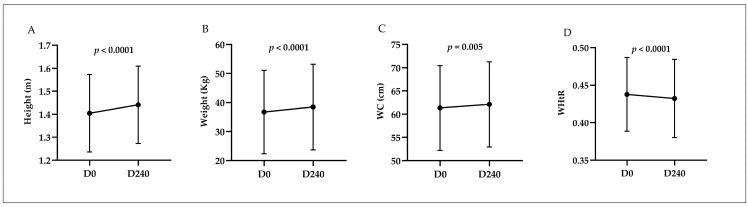
Statistically significant changes in anthropometric measures before and after intervention according to the Wilcoxon Test. D0: start of the 2019 academic year; D240: end of the 2019 academic year. WC: waist circumference; WHtR: waist-to-height ratio. Paired analysis of the measurements of: (**A**) Height (m); (**B**) Weight (Kg); (**C**) WC (cm); (**D**) WHtR.

**Figure 3 ijms-25-13073-f003:**
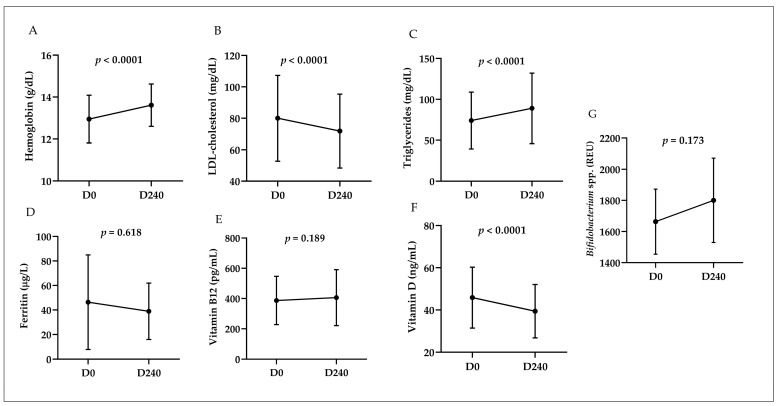
Laboratory findings pre- and post-intervention: insights from Wilcoxon test analysis. D0: start of the 2019 academic year; D240: end of the 2019 academic year. The serum vitamin D indicator evaluated was the circulating 25-hydroxyvitamin D3 (25-OH-vitamin D3). Paired analysis of laboratory markers: (**A**) Hemoglobin (g/dL); (**B**) LDL cholesterol (mg/dL); (**C**) Triglycerides (mg/dL); (**D**) Ferritin (mcg/L); (**E**) Vitamin B12 (pg/mL); (**F**) Vitamin D (ng/mL); (**G**) *Bifidobacterium* spp. (REU).

**Figure 4 ijms-25-13073-f004:**
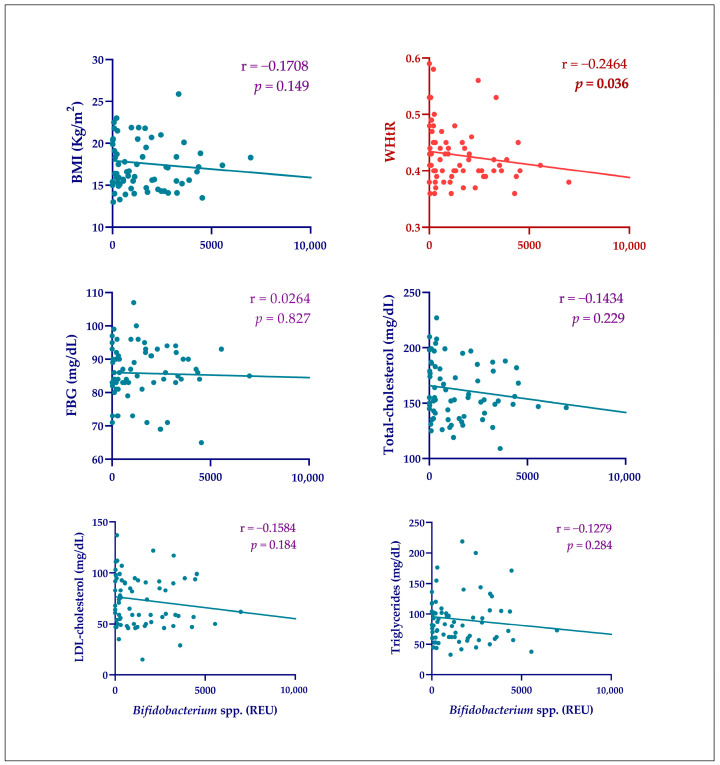
Spearman correlation between *Bifidobacterium* spp. gut abundance and clinical outcomes. D0: start of the 2019 academic year; D240: end of the 2019 academic year. BMI: Body Mass Index; WHtR: waist-to-height ratio; FBG: fasting blood glucose. Spearman correlation between WHtR and gut abundance of *Bifidobacterium* spp. (REU) highlighted in red as the only statistically significant relationship.

**Figure 5 ijms-25-13073-f005:**
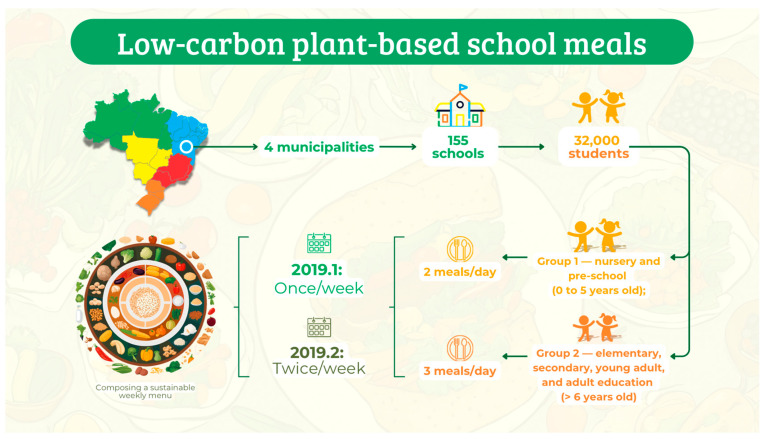
Nutritional intervention overview: replacement of animal-based and processed foods with plant-based alternatives.

**Table 1 ijms-25-13073-t001:** Frequency of laboratory findings before and after intervention in study participants.

	Laboratory Findings		
	D0 (n = 122) n (%)	D240 (n = 106) n (%)	*p* ^1^
Low hemoglobin	13 (11)	2 (2)	**0.007**
Low ferritin	8 (7)	7 (7)	1.000
Vitamin D deficiency	1 (1)	2 (2)	0.598
Vitamin B12 deficiency	19 (16)	8 (9)	0.210
Hyperglycemia	7 (6)	3 (4)	0.743
High total cholesterol	17 (14)	6 (6)	**0.049**
High LDL-cholesterol	6 (5)	1 (1)	0.127
Hypertriglyceridemia	16 (13)	24 (23)	0.079

^1^ Fisher’s exact test. *p* values in bold indicate statistically significant differences (<0.05). D0: start of the 2019 academic year; D240: end of the 2019 academic year. Reference values for laboratory parameters: low hemoglobin (5 to 12 y < 11.5 g/dL, girls aged 13 to 18 y < 12 g/dL, boys aged 13 to 18 y < 13 g/dL); low ferritin (<15 µg/L); hyperglycemia (≥100 mg/dL); high total cholesterol (≥200 mg/dL); high LDL-cholesterol (≥130 mg/dL); hypertriglyceridemia (5 to 10 y ≥ 100 mg/dL; 11 to 19 y ≥ 150 mg/dL); vitamin B12 deficiency (<200 pg/mL); vitamin D deficiency (25-OH-vitamin D3 <20 ng/mL).

**Table 2 ijms-25-13073-t002:** Generalized Estimating Equations (GEEs) model results for school meal intervention and *Bifidobacterium* spp. gut abundance effects on clinical outcomes.

Outcomes	Intervention			BIF Abundance		
	EC	SE	*p*	EC	SE	*p*
BMI	−0.090	0.311	0.772	−0.00015	0.00011	0.167
WHtR	−0.010	0.004	**0.023**	−0.00001	0.00001	0.070
Hemoglobin	0.752	0.125	**<0.001**	0.00001	0.00003	0.516
FBG	2.159	1.317	0.101	−0.00068	0.00034	**0.043**
Total Cholesterol	−0.322	3.180	0.919	−0.00181	0.00105	0.086
LDL-C	−7.488	2.836	**0.008**	−0.00203	0.00082	**0.012**
Triglycerides	17.041	5.136	**<0.001**	−0.00057	0.00120	0.631

BIF: *Bifidobacterium* spp.; BMI: Body Mass Index; WHtR: waist-to-height ratio; FBG: fasting blood glucose; LDL-C: low-density lipoprotein cholesterol; ECs: estimated coefficients; SE: standard error. *p* values in bold indicate statistically significant differences (<0.05).

## Data Availability

The dataset used to support the results presented in this study is available at the FIOCRUZ institutional data repository ARCA DADOS https://doi.org/10.35078/QO6VZB.

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
