# Peer review of "Impact of Plant-Based School Meals on Gut Bifidobacterium spp. Abundance and Health Outcomes in Schoolchildren from Bahia, Brazil"

_ijms, 2024, doi:10.3390/ijms252313073_

Round 1
Reviewer 1 Report
Comments and Suggestions for Authors
The authors investigated the effects of plant-based diets on clinical/biochemical data as well as Bifidobacterium abundance by a human study. The present study was important to considering the future nutrition in the schoolchildren. To improve the quality of this paper, the authors should revise it according to the following suggestions;
1)The abundance rate of Bifidobacteria may depend on the amount of dietary fiber ingested. Please indicate the amount of dietary fiber ingested per day. Also, please clarify the amount of energy intake before and after.
2) Quantification of the Bifidobacterium genus was performed only by PCR measurement, but the genus Bifidobacterium includes a wide variety of species. This is a limitation of this study.
3) The impact of the plant-based diet on each individual's mental state and satisfaction may have an impact. Have you conducted a questionnaire survey on such points?
Reviewer 2 Report
Comments and Suggestions for Authors
The manuscript by Camilla Almeida Menezes et al. is very relevant to bind several health factors with the composition of the school meals. As only the abundance of Bifidobacterial strains were tackled from the gut microbiota, the presented study does not give the full picture of the potential links between gut microbiota composition and health outcomes. Nevertheless, the study is carefully conducted on al alrge sample size.
The title should be rephrased as the relevance of Bifidobacteria spp. in the outcome cannot be directly concluded from the results.
Reviewer 3 Report
Comments and Suggestions for Authors
The well-documented manuscript draws attention to a very important problem: childhood obesity. The publication seeks answers to the beneficial effects of plant-based diets, weight changes and metabolic changes. But they also want to investigate how dietary changes affect the microbiome, and they want to do so by looking at Bifidobacteria spp abundance changes. One of the main disadvantages of this study is the absence of a control group.
Another drawback is the extensive and reliable study of changes in intestinal bacteria, that is, the microbiome. Unfortunately, Bifidobacteria spp. alone is not representative of changes in the microbiome. The effect of diet on the microbiome would best be demonstrated by metagenomic studies.
The examination of additional bacteria (Lacotbacillus, Ruminococcus, Akkermansia, Faecalibacterium), bacteria important for intestinal mucosal protection, even bacterial metabolites such as SCFA measurement and the examination of markers suitable for detecting leaky bowel syndrome (zonulin, LPS, FABP2, LBP, sCD14, I-FABP) may be useful.
I have the following questions about the investigation.
About 70% of the schoolchildren enrolled under the age of 5 received antibiotic treatment. How is the use of this antibiotic associated with later-developed obesity? It is known that antibiotic use in early childhood can lead to obesity later on. Of course, this may be due to altered gut microbiome diversity and leaky gut syndrome.
How can we explain why calories and nutrients intake were higher at the end of the study compared to before the study?
Could schoolchildren have been compensated at home with other types of food intake in their calorie intake because of the insufficient caloric content of plant-based nutrients?
Are there any results regarding the quality and quantity of food consumption outside school?
The decrease in vitamin D amounts can be explained by less sunshine. But the question is, what about students who do not consume products of plant origin but of animal origin?
Is this decrease also observed there, or is the rate of decrease is different?
The primary objective of this paper is to investigate the increase abundance of Bifidobacteria spp. Although the paper presents very valuable data and results on the beneficial effects of introducing a plant-based diet, it does not achieve the goal that was the primary motivator. The beneficial microbial effect of a plant-based diet cannot be studied by examining Bifidobacterium species alone. This requires a complete study of the composition of the microbiome or the examination and presentation of various metabolites.
In my view, this manuscript does not meet the expected requirements.
I recommend it for publication only after a major revision.
Round 2
Reviewer 3 Report
Comments and Suggestions for Authors
Thank you for your answers. I understand and accept the problems and financial difficulties encountered during the design and implementation of the clinical trial. Nevertheless, an appropriate control population and broader microbiome analysis could have provided very valuable work and data.
The data presented in this paper are equally valuable and groundbreaking in demonstrating the impact of healthy lifestyles on the weight of schoolchildren.
I accept the answers and support the manuscript for publication.